# Effects of an integrated mobile health lifestyle intervention among overweight and obese women planning for pregnancy in Singapore: protocol for the single-arm healthy early life moments in Singapore (HELMS) study

Jerry Kok Yen Chan,[1,2,3] Chee Wai Ku [1,2] See Ling Loy,[1,2] Keith M Godfrey,[4,5] Yiping Fan,[1] Mei Chien Chua,[2,6] Fabian Yap[2,7]

For numbered affiliations see end of article.

**Correspondence to**
Dr Jerry Kok Yen Chan;
jerrychan@duke-nus.edu.sg

## ABSTRACT

**Introduction** Changes in social and lifestyle factors have led to increasing rates of metabolic and mental health problems. We hypothesise that a transformation of the current maternal and child health system is required to deliver interventions that effectively promote a good start to life in populations at risk of metabolic and mental health problems. We describe a single-arm implementation study 'Healthy Early Life Moments in Singapore', which aims to examine whether an integrated lifestyle intervention initiated at preconception and continuing throughout pregnancy and postpartum periods can improve the metabolic and mental health of overweight and obese women, and improve early child growth.

**Methods and analysis** This single-centre implementation trial is conducted at KK Women's and Children's Hospital, Singapore. The trial aims to recruit 500 women, aged 21–40 years with a body mass index of 25–40 kg/m$^2$ who plan to get pregnant, with interventions delivered before conception, until 18 months postdelivery. Primary outcomes comprise pregnancy rate, maternal metabolic and mental health status. Secondary outcomes include maternal reproductive health, pregnancy outcomes and offspring growth. The intervention will be delivered using a mobile health application, to provide anticipatory guidance, raise awareness and guide goal-setting on lifestyle behaviours that include diet, physical activity, mental wellness and sleep hygiene from preconception to postpartum. Women who conceive within 1 year of recruitment will be followed through pregnancy and studied with their infants at six-time points during the first 18 months of life. Questionnaires, anthropometric measurements and multiple biosamples will be collected at each visit.

**Ethics and dissemination** The study has been approved by the Centralised Institutional Review Board of SingHealth (2021/2247). Written informed consent will be obtained from all participants. The findings will be published in peer-reviewed journals and disseminated to national and international policy makers.

**Trial registration number** NCT05207059.

## STRENGTHS AND LIMITATIONS OF THIS STUDY

⇒ Healthy Early Life Moments in Singapore (HELMS) aims to examine the impact of integrated interventions, initiated preconceptionally and continued throughout the pregnancy and postpartum phases, on metabolic and mental health of women who are overweight or obese.

⇒ Extensive biosampling and detailed phenotyping of mother, father and child will allow longitudinal assessment of changes in health behaviours including diet, physical activity and sleep, maternal metabolic profile and depression risk.

⇒ This will provide an important platform for biomarker discovery, validate integrated HELMS interventions and pave the way for new guidelines for lifestyle programmes from preconception to postpartum stages.

⇒ The main limitation of this study design includes the inability to distinguish between the effect of an intervention, a placebo effect and the effect of natural history.

⇒ HELMS is a single-centre study of the Asian population, involving only English-speaking participants, so external validity may be limited to this population, and caution should be exercised in generalising results across different settings and populations.

## INTRODUCTION

Non-communicable diseases (NCDs) represent the predominant cause of morbidity and mortality worldwide.[1] Rapid urbanisation, unhealthy diet and sedentary lifestyles have led to an epidemic of metabolic diseases, which are the main drivers of NCDs.[1] Coupled with this metabolic epidemic is a rising rate of mental disorders, especially depression, as the leading cause of disability worldwide that commonly affects women.[2] Both metabolic and

mental disorders are interrelated, with their co-occurrence frequently observed in individuals living with obesity.[3][4] The short-term and long-term health risks of maternal obesity and depression in mothers and their children are well documented.[5-7] Women living with obesity and mental health conditions are at increased risk of infertility, adverse obstetric outcomes and postpartum cardiometabolic complications; and their children are also susceptible to obesity, metabolic disease and psychosocial disturbances in childhood and adolescence.[5-7] In Singapore, almost one-third of women are overweight or obese (body mass index (BMI) 25–29 kg/m$^2$ :17%; BMI ≥30 kg/m$^2$ :13%) before pregnancy,[8] while one in ten women experience depression before, during and after pregnancy.[9][10]

To mitigate the impact of maternal obesity and depression, various intervention strategies targeting antenatal and postpartum periods have been studied. These include setting Specific, Measurable, Achievable, Relevant, Time-Bound goals, providing healthy lifestyle counselling or group-based education, perinatal mental health interventions and providing health service support.[11-13] However, these antenatal and immediate postpartum-phase focused interventions have had modest success,[14][15] only some showing improvements in eating behaviour, physical activity or weight status,[11][16-18] and most failing to prevent adverse mother–offspring outcomes such as gestational diabetes mellitus and macrosomia. This could be attributed to the relatively short duration of the intervention, the lack of continued care from preconception and intervention only beginning in the antenatal and postpartum periods, which may be too late to produce any meaningful impact.

There is increasing focus on interventions during the preconception period, especially for women living with obesity, who are susceptible to both metabolic and mental disorders.[3][19] Such interventions not only improve the health of the mother, but also provide potential health benefits to the next generation through an improved environment for embryo and fetal development.[20][21] The preconception period represents a unique opportunity where women are motivated to make a positive change to attain optimal pregnancy outcomes.[22] So far, preconception lifestyle interventions have demonstrated positive maternal behaviour changes, such as increased intake of multivitamins, vegetables and other dietary changes, increased physical activity, as well as smoking cessation and reduced alcohol consumption, resulting in a reduction in BMI and gestational weight gain, higher clinical pregnancy rate and lower preterm delivery.[23-25] This has led to improved knowledge, self-efficacy and control.[26] Furthermore, women who achieved at least 5% wt loss during such interventions had better cardiometabolic health based on glycaemic and lipid measures 6 years later.[25]

Taken together, this suggests that adopting a life-course perspective in the healthcare model with a continuum of care provided before, during and after pregnancy has the potential to address the trajectory of increased metabolic and mental disorder risk throughout these life stages. Acting upstream before conception to optimise health is crucial to improving reproductive potential while reducing the social and health implications resulting from unplanned pregnancies. Modifying current antenatal care services by empowering women in preparation for pregnancy and childbirth is critical for the well-being of both mother and baby. Optimising postpartum recovery and readiness, and nurturing infants through optimal feeding practices and growth monitoring are important for promoting virtuous life cycles of health in an individual and breaking vicious life cycles of NCDs for the next generation.[27]

## Goals and aims

We envision an integrated life-course approach with a continuum of care that encompasses preconception preparation, pregnancy optimisation, along with postpartum synchronisation of maternal–child health services in the first 18 months of life. Connecting the preconception, pregnancy and postpartum journeys will achieve synergism in producing greater behavioural change and beneficial outcomes not only for women, but also for the child and family.[19] Therefore, the goal of Healthy Early Life Moments in Singapore (HELMS) is to develop and implement a life-course model of care (MOC) starting from preconception to pregnancy and postpartum phases, to achieve optimal metabolic and mental health

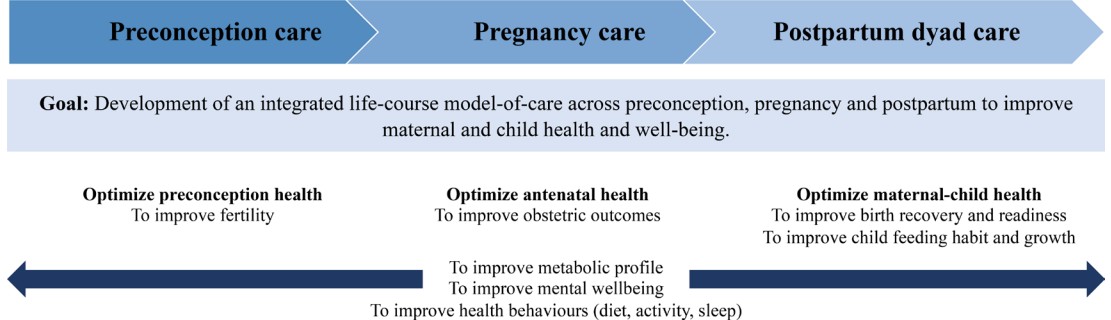

**Figure 1** The Healthy Early Life Moments in Singapore (HELMS) framework, comprising preconception, pregnancy and postpartum dyad care. Through maternal behavioural intervention, the overarching aim is to improve metabolic and mental health in women living with obesity, and to improve early child growth. Specifically, HELMS aims to improve women's fertility, obstetric outcomes, physical and mental recovery from birth, feeding habits and growth of the child.

for both mother and child. This MOC focuses on preventive healthcare, a time where the cost-effectiveness of the intervention is likely to be maximum,[28] and represents the clinical translation of early developmental programming based on the Developmental Origins of Health and Disease paradigm.[20]

In this study, the overarching aim is to examine whether an integrated intervention beginning preconceptionally and continuing throughout the pregnancy and postpartum phases can improve the metabolic and mental health of women living with obesity, as well as optimise offspring growth and development. At each life-course phase, the study addresses specific hypotheses. We hypothesise that the HELMS MOC lifestyle interventions will promote the metabolic and mental health of overweight and obese women, and thus optimise (1) reproductive outcomes during preconception, (2) obstetric outcomes during pregnancy and (3) postpartum physical and mental well-being, and healthy feeding habits and growth during infancy. We also hypothesise that greater improvements in metabolic and mental health will be observed in obese women with obesity than in those who are overweight. An overview of the study framework is shown in figure 1.

## METHODS AND ANALYSIS
### Trial design and setting
This study represents a model of implementation to demonstrate the value of adopting a life-course approach and integrated care to maternal–child health. It is conceptualised as a single-arm trial, with a targeted intervention implemented in a group of women living with obesity who are trying to conceive within 1 year after recruitment. The longitudinal intervention will be performed from preconception, throughout pregnancy until 18 months postpartum in the KK Women's and Children's Hospital (KKH), Singapore. The protocol is written following the SPIRIT (Standard Protocol Items: Recommendations for Interventional Trials) guidelines. Written informed consent will be obtained from all participants at the recruitment.

The flow of the trial is shown in figure 2. Participants will be reviewed at multiple time points from preconception through pregnancy, and both participants and their infants will be followed for the first 18 months after delivery, with extensive biosampling and detailed phenotyping performed longitudinally. Following informed consent at the first preconception clinic visit, participants will complete a set of questionnaires that assess baseline demographics, diet practices, physical activity, sleep, emotion and sexual health. Anthropometric measurements will be taken, together with biosample collection, including blood for a standard 75 g oral glucose tolerance test and stool. Participants will receive a doctor's consultation, along with prescription of a standard preconception supplement, digitised preconception care and healthy lifestyle guidance. The second preconception clinic visit will take place 6 months later, with questionnaires and

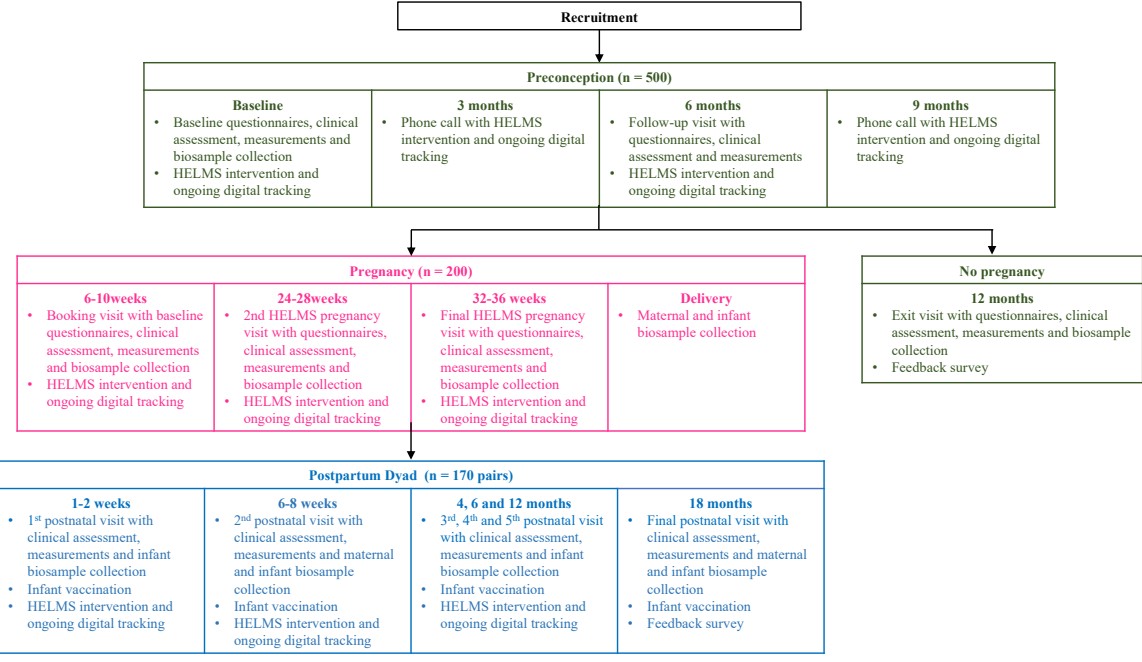

**Figure 2** HELMS study flow diagram. Preconception visits are coloured green, pregnancy visits pink and postpartum visits blue. HELMS, Healthy Early Life Moments in Singapore.

measurements being performed, and compliance with the intervention being reviewed. Phone contact will be made 3 and 9 months after recruitment to help participants maintain focus on their preconception health and lifestyle goals, and to remind them to perform a urine pregnancy test if periods are missed. If they do not become pregnant, an exit visit will be conducted at 12 months with questionnaires, measurements and biosampling, followed by appropriate evaluations and subfertility referrals thereafter. Participants who become pregnant will be seen between 6 and 10 weeks of gestation at the first pregnancy visit. Questionnaires, measurements and biosampling will again be performed. The second HELMS pregnancy visit will be at 24–28 weeks, and the final HELMS visit in pregnancy will be at 32–36 weeks. Continuous antenatal care and healthy lifestyle guidance during pregnancy will be provided throughout the trimesters. At delivery, maternal and infant biosampling, including placenta collection, will be performed. In the postpartum dyad phase, there will be six visits over an 18-month period, according to the infant vaccination schedule, at 1–2 weeks, 6–8 weeks, 4, 6, 12 and 18 months. An ongoing intervention tailored to this postpartum dyad phase will be delivered, together with questionnaires, measurements and biosampling for the mother and child at each visit. Throughout the study, participants will be asked to log their weight and supplement intake weekly in a digital calendar. For the partner, lifestyle and anthropometric measurements will be collected during the preconception, pregnancy and postpartum periods; blood will be collected once during the pregnancy phase. Meanwhile, partners will be passively engaged in the intervention by receiving the same set of digitised intervention materials. A feedback survey on the HELMS programme will be performed at the end of each preconception, pregnancy and postpartum phase.

### Patients and public involvement

We have conducted in-depth face-to-face interviews with preconception, pregnant and postpartum women who were overweight and obese to understand their needs throughout the preconception-pregnancy-postpartum journey, enablers and barriers to a healthy lifestyle, intervention preferences and delivery methods. This work enabled the development of the intervention package and dissemination plans.[29]

### Participants and recruitment

Potential participants from the public or KKH (patients/staff) or referred from other healthcare institutes who fulfil the inclusion criteria will be invited to participate in the HELMS study. KKH houses the largest Obstetrics and Gynaecology department in Singapore, managing approximately 12 000 deliveries a year with patients made up of a wide sociodemographic spectrum. Invitation letters, emails, messages, webpages, brochures and posters will be used for advertising. Interested women can contact the study team by email or phone.

Inclusion criteria include (1) a woman who plans to conceive in the next 12 months; (2) BMI 25–40 $kg/m^2$; (3) age 21–40 years; (4) Chinese, Malay, Indian ethnicity or any combination of these three ethnic groups; (5) intending to reside in Singapore for the next 4 years; (5) can access the Internet through any digital platform and (6) can provide written informed consent. Exclusion criteria include (1) currently pregnant; (2) difficulty in understanding the English language; (3) known type 1 or type 2 diabetes; (4) being on anticonvulsant medication, oral steroid, contraception or fertility medication in the past 1 month; (5) on HIV or Hepatitis B or C medication in the past 1 month. If a woman is pregnant within 1 month from the baseline visit, they will be censored from the analyses. If a woman has a pregnancy loss and wishes to rejoin the study, she will be recharacterised at the preconception baseline visit at least 1 month after a negative urine pregnancy test and will receive intervention as before.

Women who suffer from a miscarriage or termination event, experience multiple pregnancies, with fetal congenital anomalies, cannot comply with the study protocol or wish to discontinue their participation will be withdrawn from the study.

### Intervention overview

All participants will receive an intervention via a mobile health application, designed to address and improve women's knowledge, attitude and practice in terms of preconception-pregnancy-postpartum care and health behaviours. There will be four modules, namely the HELMS Journey, the HELMS Model, the HELMS Lifestyle and the HELMS Community. The HELMS Journey will provide anticipatory guidance on examinations and measurements performed during clinic visits. The HELMS Model will deliver the 4S (Screening, Size, Supplementation, Special considerations) care plans. The HELMS Lifestyle will provide support in healthy eating, physical activity, mental wellness and sleep hygiene. Finally, the HELMS Community will provide social support and improve participation in the programme. Each of these modules contains phase-specific information developed by obstetricians, neonatologists, paediatricians, dietitians, physiotherapists, psychiatrists and psychologists.

### Intervention details
#### HELMS journey

Details of each preconception, pregnancy and postpartum visit of HELMS will be available here to provide anticipatory guidance on these visits. This will ensure that participants are aware of the study measurements, as well as the required routine examinations.

#### HELMS model

The HELMS life-course interventions, namely 4S, are developed to provide care throughout the journey from preconception to postnatal. 4S is represented

by 'Screening', 'Size', 'Supplementation' and 'Special considerations'.

► Screening involves health and risk assessments through physical and biomarker (eg, Anti-Mullerian hormone, glucose, insulin and lipid profile) measurements, as well as emotion and sleep evaluations for mothers, and developmental assessment for children.

► Body Size management encompasses education on weight status awareness and mother–child health implications, both mother–child weight tracking, healthy eating and physical activity guidance.

► Supplementation includes multimicronutrient, vitamin D, calcium and/or DHA supplements, which are phase specific for mother, and vitamin D supplement drops for the child.

► Special considerations include the management of preconception sexual health and function, the management of pregnancy symptoms and postpartum recovery, and the monitoring of infant growth and feeding management.

### HELMS lifestyle

Metabolic health support (diet and physical activity modules).

The diet module is developed based on the 6P tool, which is designed as a platform to address metabolic health based on the principles of energy input and expenditure, as well as motivation as the basis for dietary change.[30] It is a simple, self-administered instrument based on the concept of a mental model, which can lead to self-awareness, self-evaluation and self-education, over time resulting in a positive change in eating habits and health. The 6P comprises six components as presented in table 1. Multiple modalities are incorporated that include feedback on 6P behaviour, 6P self-monitoring, 6P goal setting and 6P nudges along with the use of the 6P tool. By including these six main discrete dietary and activity components under a single 'package', it will provide an overview of the mental model of nutrition based on the principle of energy input and expenditure, promoting self-awareness of unhealthy lifestyle behaviours and nudging individuals into actual implementation via concrete, personalised feedback and increasing intrinsic motivation. This is based on the theoretical framework of the theory of planned behaviour, which has been widely used to predict behavioural intention and health-related behaviours.[31]

The physical activity module seeks to engage women with the importance, type and intensity of exercise during each phase. It consists of a series of appropriate, tailored exercise videos developed by physiotherapists and obstetricians with the support of scientific evidence. Wearable activity trackers will be used to provide real-time feedback on physical performance and help participants monitor their energy expenditure. Sustainability will be supported with the use of nudges (both text messages and images) delivered weekly to influence behaviour and decision-making.

### Mental health support (Mental health and Sleep modules)

The mental health module serves as the main way to support, screen and guide women's mental health longitudinally. Upstream screening of depressive symptoms at the preconception stage allows women at high risk to be triaged earlier for receiving targeted mental health support from KKH's mental health team comprising a clinical coordinator, case manager and psychiatrist. Women at medium risk will be engaged more intensively, while those at low risk will continue to participate, enabling and preparing for different stages through a package of clinical care and lifestyle support, especially in the late stages of pregnancy and early stages of postpartum. The sleep module aims to support mental wellness by promoting healthy sleep practices throughout the life-course. A wearable sleep tracker will be used to provide real-time feedback on sleep performance and help participants to monitor their sleep patterns.

### HELMS community

An online community platform is being established as part of supporting mental health among HELMS participants. This allows women to share their experiences, identify common problems among peers and generate solutions that work for them in the online chat group.

Our intervention approach will be based on the SIGN strategy (Support, Inform, Guide and Nudge) to deliver the 4S and healthy lifestyle support. Close monitoring of women's health allows timely support provided by the healthcare staff. Setting up mobile health education helps to inform women about the importance and warning points in different stages. Educational materials in various formats, including diagrams and videos, provide guidance on phase-specific care and healthy lifestyles that focus on nutrition, physical activity, mental health and sleep hygiene. Health nudges in the form of key messages related to preconception, pregnancy and postpartum dyad care, as well as healthy lifestyles, facilitate care support and behavioural changes. The use of nudges represents a preferred architecture strategy that is widely used in public policy making, to alter people's behaviour and influence decision making.[32]

| Table 1 | Components of the 6P tool | |
|---------|------------------------------|------------|
| 6P | Healthy mental model of nutrition | Description |
| P1 | Portion | Amount of food intake |
| P2 | Proportion | Caloric density of food intake |
| P3 | Pleasure | Frequency of snacks and sugary beverages and irregular intake |
| P4 | Period | Time of day of food intake |
| P5 | Physicality | Physical activity and sedentary behaviour |
| P6 | Psychology | Readiness for change |

In summary, we will apply an interactive and personalised approach to conducting the intervention, supported by real-time feedback. This involves goal setting, individual education, self-awareness, self-monitoring, motivation and outcome review process. These techniques are commonly used in behavioural interventions and can improve health outcomes.[33 34]

### Intervention adherence

Adherence to the intervention will be established throughout the study with regular logbook charts of supplement intake, sexual activity and weight, which will be reviewed at each study visit or by telephone reminders. The use of a digital tracker for activity and sleep allows offsite monitoring of wearing adherence, and research staff will take action to check for reasons for non-adherence.

### Outcomes and assessments

The primary outcome is the pregnancy rate in overweight and obese women trying to conceive within 1 year of baseline assessment. It is defined by a positive urine pregnancy test, followed by ultrasound confirmation of an intrauterine gestational sac after 6 weeks of amenorrhoea. If an ultrasound scan is not available or inconclusive, the diagnosis of pregnancy will be made clinically. A successful conception is one of the most important clinical outcomes, and it represents the culmination of improved metabolic and mental health, leading to optimised gametogenesis.

Coprimary outcomes include maternal metabolic health and mental health status in each phase, with 18 months postdelivery serving as the principal endpoint. Maternal metabolic health is assessed by metabolic syndrome criteria. Mental health is evaluated using the Edinburgh Postnatal Depression Scale (EPDS).[35]

Other key outcomes include:

#### Women/mothers

► Reproductive health as assessed by fecundability, pregnancy loss and live birth rates, and sexual function based on Female Sexual Function Index-19.[36]
► Pregnancy outcomes include pain, obstetric and delivery complications.
► Health behaviours assessed by dietary practice (6P tool,[30] 4-day food diary, Food Frequency Questionnaire,[37] Three Factor Eating Questionnaire,[38] alcohol and supplement intake), smoking exposure, physical activity and sedentary behaviours (International Physical Activity Questionnaire,[39 40] Sedentary Behaviour Questionnaire,[41] accelerometer), and sleep (Pittsburgh Sleep Quality Index,[42] digital tracker).
► Metabolic profile, including weight status, body fat distribution, blood pressure, lipid and glycaemic measures.
► Gut microbiome profile assessed by stool sample.
► Nutrient status based on diet, blood and breast milk composition (breast milk will be collected at five time points through the first year postdelivery).
► Ocular health as assessed by retinal vessel calibre characteristics.

#### Offspring

► Anthropometry and growth assessed by antenatal serial ultrasound scans, infant weight, length and head circumference.
► Neonatal complications such as hypoglycaemia and admission to neonatal care facilities; infant health and well-being, and skin biopsy for a subset of infants with eczema.
► Infant feeding assessed by breastfeeding behaviours, time of weaning, nutrition milestone, dietary intake and eating behaviours.

#### Partners

► Physical and metabolic health assessed by weight, BMI and body fat distribution.
► Health behaviours assessed by smoking exposure, alcohol intake, meal pattern, stress level, physical activity and sedentary behaviours.

Programme effectiveness will be evaluated based on women's quality of life, healthcare utilisation and participant feedback surveys. Blood, cord blood, stool, urine, saliva, placenta and breast milk samples will be stored for analyses on biochemical, micronutrient, metabolomic, genomic, epigenetic, immunological and molecular profiles.

### Planned analyses

Continuous variables will be presented as means and SD, or medians and 25th–75th centiles, as appropriate. Categorical variables will be presented as numbers and percentages. The pregnancy rate will be determined by the number of women who became pregnant (defined as stated previously) divided by the total number of women who completed the 1-year follow-up during the preconception phase. The time to pregnancy (TTP) will be estimated by the number of menstrual cycles required to achieve pregnancy over 1 year of follow-up. We will use the discrete-time proportional hazards model, which analyses TTP as a discrete scale based on the number of menstrual cycles, to estimate the fecundability ratio (FR) and the 95% CI,[43 44] accounting for left truncation and right censoring. For the other two coprimary outcomes, a linear mixed effects model will be used to examine changes in means between the baseline and follow-up metabolic markers and EPDS scores, with adjustment for baseline potential confounders (including partner's characteristics) and duration of the intervention received. In addition to the pre–post comparison of outcomes stated above, differences in pregnancy and birth outcomes will also be compared with a similar observational cohort in Singapore called Singapore PREconception Study of long-Term maternal and child Outcomes (S-PRESTO),[45] using multiple linear or logistic regression models, adjusting for potential confounders. These confounders, such as age, ethnicity and education, will be determined from

literature review, directed acyclic graph and/or observed statistically significant associations with exposures and outcomes.

We will impute missing data using multiple imputation analyses by chained equations.[46] The number of imputations will be determined based on a percentage of missing values,[47] and the results of total imputations will be pooled using Rubin's rule.[48] To determine whether the imputation of the missing data may have affected the results, we will perform sensitivity analyses on participants with a complete set of data.

### Sample size calculation

Based on the S-PRESTO study,[45] which is an observational cohort recruited from preconception and followed up through postpartum periods, preconception overweight or obese women (BMI ≥25 kg/m$^2$) had a 38% pregnancy rate, while those with normal weight (BMI 18.5–24.9) had 47% pregnancy rate. Among overweight and obese women, those with healthy metabolic profiles characterised by absence of the metabolic syndrome or insulin resistance had a 52% pregnancy rate. We anticipated that HELMS intervention will improve overweight or obese women's pregnancy rate from 38% to 50% within 12 months of trying. Using a two-sided α level of 5% and a power of at least 90%, 400 participants are needed in the preconception phase, leading to 200 pregnancies. Considering a 20% drop-out rate, 500 participants are required for recruitment. We estimated that 15% of women will experience pregnancy loss and another 10% drop-out, leaving 150 dyad pairs to be followed.

### Quality control

Procedures and actions will be implemented throughout the study to ensure that information provided to all participants is standardised, data collected is as complete as possible and of high quality. Research staff responsible for recruiting and follow-up participants will receive training from study leads on recruitment, consent taking, questionnaire and data management, intervention delivery and compliance monitoring. An operation database will be developed to monitor participant progression throughout the study, schedule study visits and monitor visit/measure completeness. A quarterly meeting will be held among study leads and research staff to review recruitment process, intervention delivery, data collection and participants' feedback. An annual audit on the study will be performed by an independent party.

### Data monitoring

All data will be pseudonymised. Participants' identifiers will be kept separately in a password-protected file and only be assessed by specific research staff. Electronic data will be managed using a secure, encrypted online data-capturing system approved by the institution. A data monitoring team will perform data checking on completeness, errors and outliers. To facilitate data monitoring procedure, algorithms are programmed to autodetect implausible value while entering data and to prompt messages for incomplete entry. Ad hoc discussions will be organised with study leads to clarify actions required to resolve data issues.

### ETHICS AND DISSEMINATION

The study has been approved by the Centralised Institutional Review Board of SingHealth (2021/2247). Written informed consent will be obtained from all participants (online supplemental file 1). The findings will be published in peer-reviewed journals and disseminated to national and international policy makers.

### DISCUSSION

The existing MOC in maternal–child health in Singapore and in many developed countries emphasises disease treatment with less emphasis on health promotion. Most women are not prepared for pregnancy, and many have poor metabolic and mental health. This lack of preparedness carries over in the postpartum phase after the child is born. A woman's own needs are often neglected in the immediate postpartum phase when the attention is primarily focused on caring for the newborn. Coupled with unexpected physical and emotional challenges in this new phase of life, it culminates in a cycle of poor metabolic and mental health. Women living with overweight and obesity represent a particularly high-risk group for poor metabolic and mental health.[49 50] The current obesity management strategy has three fundamental gaps. First, the timing of intervention is too far downstream in adulthood. Second, there is weak foundational knowledge of nutrition and physical activity, with poor awareness and insight into metabolic health. Third, there is no structured and coordinated care plan for these women while planning to conceive, which would be an ideal window for intervention during the crucial period of gametogenesis.[51] To address these challenges, there is a need to invert the pyramid of care and focus our effects upstream, starting from preconception.

HELMS aims to address the twin challenges of metabolic and mental health disorders through a series of interventions, to provide the best and most equal start to life for children. Through its life-course approach, HELMS recognises the additive effects of influences on a person's life that will impact the future health trajectory. It aims to impart new mental models of nutrition, physicality and mental wellness, by guiding, supporting and empowering women through an integrated journey from preconception to postpartum, to address metabolic and mental health challenges. We hope this unique holistic approach with a continuum of care across the early life-course, starting from preconception and pregnancy until the postpartum period will build a strong foundation for healthy women and secure a healthy start for future generations.

Drawing on the experience and evidence derived from two local prospective observational mother–offspring cohorts conducted at KKH and National University Hospital (NUH) since 2008—Growing Up in Singapore Toward Healthy Outcomes (GUSTO) and the S-PRESTO,[45 52] HELMS seek to examine the impact of integrated interventions, initiated preconceptionally, on metabolic and mental health of women living with overweight or obesity. Extensive biosampling and detailed phenotyping of mother, father and child will allow longitudinal assessment of changes in health behaviours including diet, physical activity and sleep, maternal metabolic profile and depression risk. The biosamples will provide an important platform for biomarker discovery, validate integrated HELMS interventions and pave the way for new guidelines for lifestyle programmes from the preconception to postpartum stages.

Despite the strong evidence base for adverse maternal and child outcomes in overweight and obese women, as well as the benefit of lifestyle interventions in the preconception and pregnancy phases, integrated HELMS interventions are novel and have not established unequivocal evidence of benefit. Thus, HELMS is conceptualised as a single-arm intervention trial in women living with overweight or obesity to obtain preliminary evidence of the efficacy of intervention. A subsequent larger pragmatic trial is envisaged to validate the efficacy of the intervention. The main limitations of this study design include the inability to distinguish between the effect of intervention, a placebo effect and the effect of natural history, including regression to the mean. Given the similarity in study design to the prospective observational cohort S-PRESTO, with similar time points of follow-up and biosampling, there is the potential for comparison with this historical cohort. HELMS is a single-centre study of the Asian population, involving only English-speaking participants, thus, the external validity may be limited to this population, and caution will need to be exercised in generalising results across different settings and populations. Finally, the interventions are tailored to the woman/mother, with only passive intervention for her partner. The environment of the woman plays an important role in her compliance to lifestyle interventions,[53] which may undermine the impact of our programme. On the other hand, if the success of the HELMS interventions extends to that of her immediate family, we would also have limited insights with minimal measurements on the partner, to reduce the burden of involvement in the study.

The major strength of this study is the extensive biosampling and detailed phenotyping of mother, father and child, which will allow longitudinal assessment of changes in health behaviours including diet, physical activity and sleep, maternal metabolic profile and depression risk. This provides an important platform for biomarker discovery, validation of integrated HELMS interventions and paves the way for new guidelines for lifestyle programmes throughout the life-course. We acknowledge the main limitation of this single-arm trial, where we are unable to distinguish between the effect of intervention, a placebo effect or the effect of natural history. In addition, HELMS is conducted in a developed country with only English-speaking Asian participants, since English is the language used in our assessment tools and advisories, which may limit its external validity and caution should be exercised before generalising our findings to other settings and population.

## Trial status
HELMS recruitment commenced in April 2022 and is expected to be completed in June 2024. The current protocol is version 1, dated 16 December 2021.

**Author affiliations**
[1]Department of Reproductive Medicine, KK Women's and Children's Hospital, Singapore
[2]Duke-NUS Medical School, Singapore
[3]NUS Yong Loo Lin School of Medicine, Singapore
[4]MRC Lifecourse Epidemiology Unit, University of Southampton, Southampton, UK
[5]National Institute for Health Research Southampton Biomedical Research Centre, University of Southampton, Southampton, UK
[6]Department of Neonatology, KK Women's and Children's Hospital, Singapore
[7]Department of Paediatrics, KK Women's and Children's Hospital, Singapore

**Acknowledgements** The authors would like to thank KK Women's and Children's Hospital for the institutional support received during this study. We would also like our multidisciplinary team of collaborators including, but not limited to, Dr Helen Chen, Dr Han Wee Meng, Dr Mary Chong, Dr Jacinth Tan, Dr Saumya Jamuar, Dr Salvatore Albani, Dr Joanne Yoong, Dr Ng Wei Yen, Dr Jeremy Lim, Dr Michael Chee and Dr Falk Mueller-Riemen-Schneider. The authors also thank the contributions of Ms Ng Xiang Wen, Ms Lai Lan Tian, Ms Rachael Loo Si Xuan, and Ms Faith Liew Hui Hua for their administrative support.

**Contributors** Conceptualisation, JC, FY, CWK and SLL; methodology, JC, FY, KG, CWK and SLL; writing—original draft preparation, CWK, SLL and YF; writing—review and editing, all authors; funding acquisition, MCC, FY and JC. All authors have read and agreed to the published version of the manuscript.

**Funding** This research was supported by the KKH Health Services Model of Care Transformation Fund (MoCTF) Grant (MoCTF/01/2020, MoCTF/02/2020 and MoCTF/03/2020) and the Lien Foundation Optimising Maternal and Child Health Programme Fund (grant number not applicable). CWK and JC are supported by the National Medical Research Council, Ministry of Health, Singapore (NMRC/MOH-000596-00 and NMRC/CSA-SI-008-2016, respectively). KG is supported by the National Institute for Health Research (NIHR Senior Investigator (NF-SI-0515-10042), NIHR Southampton 1000DaysPlus Global Nutrition Research Group (17/63/154) and NIHR Southampton Biomedical Research Center (IS-BRC-1215-20004), British Heart Foundation (RG/15/17/3174) and the European Union (Erasmus+ Programme ImpENSA 598488-EPP-1-2018-1-DE-EPPKA2-CBHE-JP).

**Disclaimer** The funding body did not influence either the data collection and analysis or the writing and the decision to submit the manuscript.

**Competing interests** KG and YF received reimbursement for speaking at conferences sponsored by companies that sell nutritional products. KG is part of an academic consortium that received research funding from Abbott Nutrition, Nestle and Danone. All other authors declare no competing interests.

**Patient and public involvement** Patients and/or the public were involved in the design, or conduct, or reporting, or dissemination plans of this research. Refer to the Methods section for further details.

**Patient consent for publication** Not applicable.

**Provenance and peer review** Not commissioned; externally peer reviewed.

of the translations (including but not limited to local regulations, clinical guidelines, terminology, drug names and drug dosages), and is not responsible for any error and/or omissions arising from translation and adaptation or otherwise.

**ORCID iD**
Chee Wai Ku http://orcid.org/0000-0003-3719-3005

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
