## [Reviewer comments · BMJ Open]

ARTICLE DETAILS

TITLE (PROVISIONAL)	Effects of an integrated mobile health lifestyle intervention among overweight and obese women planning for pregnancy in Singapore: protocol for the single-arm Healthy Early Life Moments in Singapore (HELMS) study
AUTHORS	Chan, Jerry; Ku, Chee Wai; Loy, See Ling; Godfrey, Keith; Fan, Yiping; Chua, Mei Chien; Yap, Fabian

VERSION 1 – REVIEW

REVIEWER	Tran, Thach Monash University, Global and Women's Health, School of Public Health and Preventive Medicine
REVIEW RETURNED	20-Mar-2022

GENERAL COMMENTS	This is a protocol of a pilot study. The information about the intervention was provided in detail which helps the readers understand the intervention well. However, my major concern is that this is a pilot study, but the aims are the aims of a main trial evaluating the effects of the intervention. If this is a pilot study, the aims should be assessing the feasibility and acceptability of the intervention and various aspects of the main trial design. With one arm, the results of this study would not be able to identify if the intervention can improve the outcomes. The sample size is quite large for a pilot. From my point of view, this protocol should be re-written with the aims and the methods of a pilot study.
--

REVIEWER	Reynolds, Leryn Old Dominion University
REVIEW RETURNED	18-Jul-2022

GENERAL COMMENTS	This is an interesting and novel study examining how lifestyle interventions delivered via a digital platform preconception, during pregnancy and for up to 18 months after delivery impact maternal and fetal health. The authors should be applauded for this tremendous undertaking. I have listed a few comments below for consideration for the authors. 1) The dates of the study are missing and this is a requirement of the journal.2) Description of how "improving fertility during preconception" is missing and given that this is a specific goal of the study would be important to elaborate on. Along these same lines, roughly 1 in 3 pregnancies ends in miscarriages. How will this be taken into consideration in terms of "improving fertility"? Will women who have a miscarriage(s) and do not have a viable pregnancy at the end of the year be considered with impaired fertility versus women who have a miscarriage but within the year go on to have a viable
--

	pregnancy? Given that a decent number of pregnancies end in miscarriage, this should be thought through before beginning. 3) It is recommended that partners have their data collected at a defined period during the study rather than “collected once throughout the entire study period” given that from preconception to 18 months post conception is a wide timeframe and it is likely that the addition of a new child in the family will impact not only maternal behaviors/lifestyle but partner behavior/lifestyles as well. 4) Exclusionary criteria are missing from the protocol. 5) What timeframe will breast milk samples be collected? The composition of breast milk changes greatly over time. 6) Inclusion criteria begins at BMI of 25.0 kg/m², though language throughout the rest of the protocol suggests populations with obesity are the target. Is there a justification for including overweight populations as well? 7) The population is limited to only English speakers in Singapore, which is mentioned as a known limitation but there is no justification written for limiting to English speakers only. A justification may prove useful. 8) Some of the aspects of the HELMS protocols appear to have been validated previously in literature, but some do not have any accompanying cited studies validating their use. Has the SIGN strategy been used in the past or validated previously? 9) If the health measures of the partners are not part of the outcomes of the study, is there a justification as to the inclusion of their measurements?
--	---

REVIEWER	Sparks, Joshua Pennington Biomedical Research Center
REVIEW RETURNED	26-Sep-2022

GENERAL COMMENTS	First, I would like to thank the Editorial Board for recommending me to review the manuscript entitled: Healthy Early Life Moments in Singapore (HELMS): Study Protocol for a Single-Arm Pilot Implementation Model. Second, I commend the authors for an exemplary job developing, testing, and implementing a trial tackling the life-course model-of-care from the preconception period, throughout pregnancy, and following up into the postpartum period to evaluate maternal metabolic and mental health, while also acquiring data about the offspring and the partner. Although the authors did a thorough job of explaining the HELMS study, there are points of concern that require revisions to the current manuscript. Please refer to the point-by-point comments for major/minor concerns, as well as suggestions for improvement. There are two separate paragraphs with goals, which equate to five (5) goals of the study. You have goals for the study itself and goals for the different time periods. This may cause confusion as to what the primary goals of the program may be. Do you have any a priori hypotheses? There are quite a few measurements and time points, but no hypotheses stated. What is the justification or calling this a pilot trial and why no comparator group, such as standard care? Recommend if you are going to highlight collection of paternal/father data to justify why you are doing so. In-line with this, you need to speak to altering the household environment. If you are targeting maternal/child health and recruiting the partner/father, it is critical to
---

	 speak to how the household environment influences preconception, pregnancy, and postpartum outcomes. Additionally, you state you are passively targeting the partner/father in the intervention, but are you accounting for actual involvement, such as collection of actual engagement (number of contacts, etc...). Lastly, when it comes to the partner/father, why don't you justify only 1 collection time point for the partner/father? There is the potential for the partner/father to be followed for 40 months during the preconception (12 months), pregnancy (10 months), and postpartum (18 months) periods. Do you not anticipate any changes to the partner/father during that time?   25% are overweight or obese in Singapore. Do you know the prevalence estimates of overweight and obese independently? Should state if so and justify how you anticipate your outcomes to differ in those having overweight versus those having obesity. Another note is that 25% is a low proportion of the population compared to other countries (US, UK, Australia, for example). Do you really anticipate recruitment of 500 women in preconception time period over 5 years? Especially since, I assume, these women need to be recruited by year 3 to allow for proper follow-up by year 5.   Similar to comment above, you plan to recruit 500 women to enroll 400 women in the preconception phase. You anticipate 50% to become pregnant because a previous trial increased pregnancy prevalence, which is a sound rationale. 200 women will be followed throughout pregnancy and 170 through the postpartum period, but at no point do you all talk about drop-out for any reason, especially since you are following these women for such a long period of time. You only account for medical drop out, which is not acceptable. You must speak to the number of anticipated women to not complete the follow-up periods, specifically through the 18-month postpartum period.   I do not necessarily agree with your statement: The preconception period represents a unique opportunity where women have the greatest motivation to make a positive change (reference 22). I actually challenge this statement unless further justification may be provided. This statement may only be true in women trying to conceive who are also actively concerned about their pregnancy. Previous evidence suggests that pregnancy is the most potent teachable moment for health behavior change. Not the planning period prior to pregnancy, unless they are actively planning. As such, I believe clarity is needed.   You discuss adherence metrics, but do not specify exact primary adherence metric for each time point or period of time. Is there a target behavior you all believe is most important? You should clearly describe your primary and secondary adherence metrics.   Is there a minimum amount of time the participant must be enrolled in the preconception period for it to be included in analysis. As an example, if someone were to enroll in preconception and 2 weeks later had a positive pregnancy test, how do you handle their preconception data?   At what time point is your criteria for primary and secondary outcomes? You have three (3) distinct time periods, each of critical importance for preconception, pregnancy, and postpartum metabolic 
--	--

	and mental health. Is it the 18-month postpartum visit you think is most important? Is so or if not, there needs to be some justification. Your discussion in the subsection "Patients and Public Involvement" about the formative work performed prior to implementation of HELMS should be highlighted much earlier. Citizen and implementation science is extremely important and a key strength of this trial. What is the digital platform utilized? A little more information would once again be very helpful. No discussion of strengths and limitations. Recommend reading other protocol papers for insight into how to improve the current protocol paper, including, but not limited to, https://doi.org/10.1186/s13063-021-05204-w; doi: 10.2196/18211; DOI: 10.1186/s13063-020-4062-4
--	---

VERSION 1 – AUTHOR RESPONSE

Reviewer 1

1. This is a protocol of a pilot study. The information about the intervention was provided in detail which helps the readers understand the intervention well. However, my major concern is that this is a pilot study, but the aims are the aims of a main trial evaluating the effects of the intervention. If this is a pilot study, the aims should be assessing the feasibility and acceptability of the intervention and various aspects of the main trial design. With one arm, the results of this study would not be able to identify if the intervention can improve the outcomes. The sample size is quite large for a pilot. From my point of view, this protocol should be re-written with the aims and the methods of a pilot study.

Response: Thank you for your comment. Although a single-arm trial, comparison of clinical outcomes with the recent similar observational cohort in Singapore called Singapore PREconception Study of long-Term maternal and child Outcomes (S-PRESTO) will allow the evaluation of the intervention effectiveness (Line 417). However, we do acknowledge that the use of the term 'pilot' is not appropriate in the present context based on the designed aims and planned analyses. This issue has been similarly pointed out by the editor in #2 above. We have removed the word 'pilot' throughout the manuscript.

Reviewer 2

This is an interesting and novel study examining how lifestyle interventions delivered via a digital platform preconception, during pregnancy and for up to 18 months after delivery impact maternal and fetal health. The authors should be applauded for this tremendous undertaking. I have listed a few comments below for consideration for the authors.

1. The dates of the study are missing and this is a requirement of the journal.

Response: We have updated the study dates under Trial Status; recruitment commenced in April 2022 and is expected to be completed in June 2024.

Lines 543-544: HELMS recruitment commenced in April 2022 and is expected to be completed in June 2024. The current protocol is version 1, dated 16 Dec 2021.

2. Description of how "improving fertility during preconception" is missing and given that this is a specific goal of the study would be important to elaborate on. Along these same lines, roughly 1 in 3 pregnancies ends in miscarriages. How will this be taken into consideration in terms of "improving fertility"? Will women who have a miscarriage(s) and do not have a viable pregnancy at the end of the year be considered with impaired fertility versus women who have a miscarriage but within the year

go on to have a viable pregnancy? Given that a decent number of pregnancies end in miscarriage, this should be thought through before beginning.

Response: The intervention aims to improve metabolic and mental health, and as a result, improve fertility. Specifically, the 4S interventions under the HELMS Model address issue of “Sex” as the fourth S which provides advice to the couple aimed to improve fertility.

The primary outcome is pregnancy rate (rather live birth rate), with pregnancy is defined by a positive urinary pregnancy test, followed by ultrasound confirmation of an intrauterine gestational sac after six weeks of amenorrhea. If an ultrasound scan is not available or inconclusive, the diagnosis of pregnancy will be made clinically. If a woman has a miscarriage, she will be withdrawn from the study, and if she wishes to re-join the study, she will be re-characterized as at the preconception baseline visit at least a month after a negative urine pregnancy test. Thus, despite a miscarriage, these women will contribute towards our primary outcome of pregnancy rate, however, the miscarriage incidence will alter the live birth rate, which is a secondary outcome in this trial. We can see how the term fertility may be misleading, and hence we have decided to change the term used to reproductive outcomes during preconception instead.

Lines 156-160: We hypothesize that the HELMS MOC lifestyle interventions will promote metabolic and mental health of overweight and obese women, and thus optimize (i) **reproductive outcomes during preconception**, (ii) obstetric outcomes during pregnancy, and (iii) postpartum physical and mental wellbeing, and healthy feeding habits and growth during infancy.

3. It is recommended that partners have their data collected at a defined period during the study rather than “collected once throughout the entire study period” given that from preconception to 18 months post conception is a wide timeframe and it is likely that the addition of a new child in the family will impact not only maternal behaviors/lifestyle but partner behavior/lifestyles as well.

Response: We have amended the text to make it clearer that we will be collecting both lifestyle and anthropometric measurements during every phase of the study, including preconception, pregnancy and postpartum periods. However, given that partners are not the main intervention target in HELMS, we have decided to collect blood only once during the pregnancy phase in order to minimize participant burden.

Lines 202-204: For the partner, lifestyle and anthropometric measurements will be collected during the preconception, pregnancy, and postpartum periods; blood will be collected once during the pregnancy phase.

4. Exclusionary criteria are missing from the protocol.

Response: Exclusion criteria are detailed in the Participants and recruitment section.

Lines 229-232: Exclusion criteria include (1) currently pregnant; (2) difficulty in understanding the English language; (3) known Type 1 or Type 2 diabetes; (4) on anticonvulsant medication, oral steroid, contraception, or fertility medication in the past one month; (5) on HIV or Hepatitis B or C medication in the past one month.

5. What timeframe will breast milk samples be collected? The composition of breast milk changes greatly over time.

Response: Thank you for the comment. Indeed, breast milk composition does change over time, and hence, we have added the timeframe of breast milk collection under the Outcomes and Assessments section.

Lines 378-379: Nutrient status based on diet, blood, and breast milk composition (breast milk will be collected at five timepoints through the first year post-delivery).

6. Inclusion criteria begins at BMI of 25.0 kg/m², though language throughout the rest of the protocol suggests populations with obesity are the target. Is there a justification for including overweight populations as well?

Response: Women who are overweight are more likely to progress to obesity if left uncared for, thus it is important for any preventive health programs to address this high risk group at the same time.

7. The population is limited to only English speakers in Singapore, which is mentioned as a known limitation but there is no justification written for limiting to English speakers only. A justification may prove useful.

Response: Our assessment tools and advisories are available only in English, underpinning exclusion of women with limited command of the language as this will affect their ability to comprehend the education materials. Based on Singapore's Census of Population 2020, 95-98.7% of our resident population between the age of 25-44 years old are English literate therefore while it is a limitation, there will be less than 5% of the population that will be excluded due to the language used. Based on the same reason, there is little justification to develop tools and advisories in another language at this point.

Lines 537-539: In addition, HELMS is conducted in a developed Asian country, and only included participants who are English-speaking, since English is the language used in our assessment tools and advisories.

Reference:

<https://www.singstat.gov.sg/-/media/files/publications/cop2020/sr1/cop2020sr1.pdf>

8. Some of the aspects of the HELMS protocols appear to have been validated previously in literature, but some do not have any accompanying cited studies validating their use. Has the SIGN strategy been used in the past or validated previously?

Response: SIGN is the communication strategy that is meant to deliver the 4S and healthy lifestyle support. It has not been validated before specifically, but each of these components are commonly used to deliver lifestyle intervention programs.

9. If the health measures of the partners are not part of the outcomes of the study, is there a justification as to the inclusion of their measurements?

Response: We acknowledge the important role of partners in the reproductive and obstetric outcomes of the women, thus, partner health measures will be used as potential confounders in our analysis. We have added a line in Planned Analyses to highlight this. Examination of changes in partner variables may inform a process evaluation of the effects of the intervention.

Lines 411-414: For the other two co-primary outcomes, linear mixed effects model will be used to examine changes in means between the baseline and follow-up metabolic markers and EPDS scores, with adjustment for baseline potential confounders (including partner's characteristics) and duration of the intervention received.

Reviewer 3

First, I would like to thank the Editorial Board for recommending me to review the manuscript entitled: Healthy Early Life Moments in Singapore (HELMS): Study Protocol for a Single-Arm Pilot Implementation Model. Second, I commend the authors for an exemplary job developing, testing, and implementing a trial tackling the life-course model-of-care from the preconception period, throughout pregnancy, and following up into the postpartum period to evaluate maternal metabolic and mental health, while also acquiring data about the offspring and the partner. Although the authors did a thorough job of explaining the HELMS study, there are points of concern that require revisions to the current manuscript. Please refer to the point-by-point comments for major/minor concerns, as well as suggestions for improvement.

1. There are two separate paragraphs with goals, which equate to five (5) goals of the study. You have goals for the study itself and goals for the different time periods. This may cause confusion as to what the primary goals of the program may be.

Do you have any a priori hypotheses? There are quite a few measurements and time points, but no hypotheses stated.

Response: Thank you for your comment, we do see how this can be confusing for the readers. Hence, we have edited the Goal and Aims section to reflect the goal and a priori hypotheses more clearly.

Lines 144-150: Therefore, the goal of Healthy Early Life Moments in Singapore (HELMS) is to develop and implement a life-course model of care (MOC) starting from preconception to pregnancy and postpartum phases, to achieve optimal metabolic and mental health for both mother and child. This MOC focuses on preventive healthcare, a time where the cost effectiveness of intervention is likely to be maximum,²⁸ and represents the clinical translation of early developmental programming based on the Developmental Origins of Health and Disease (DOHaD) paradigm.²⁰

Lines 155-160: At each life-course phase, the study addresses specific hypotheses. We hypothesize that the HELMS MOC lifestyle interventions will promote metabolic and mental health of overweight and obese women, and thus optimize (i) reproductive outcomes during preconception, (ii) obstetric outcomes during pregnancy, and (iii) postpartum physical and mental wellbeing, and healthy feeding habits and growth during infancy.

2. What is the justification or calling this a pilot trial and why no comparator group, such as standard care?

Response: We acknowledge it is not appropriate to call this study as 'pilot' and have removed the word 'pilot' throughout the paper. We have similarly addressed this as commented by the editor at #2 and reviewer 1 at #1 above. At present, there is no existing preconception and postpartum dyad standard care services in Singapore that can serve as the comparator group. Thus, this new MOC is designed as a single-arm trial, and planned to compare the clinical outcomes with a similar observational cohort in Singapore called Singapore PREconception Study of long-Term maternal and child Outcomes (S-PRESTO) (Line 417).

3. Recommend if you are going to highlight collection of paternal/father data to justify why you are doing so. In-line with this, you need to speak to altering the household environment. If you are targeting maternal/child health and recruiting the partner/father, it is critical to speak to how the household environment influences preconception, pregnancy, and postpartum outcomes. Additionally, you state you are passively targeting the partner/father in the intervention, but are you accounting for actual involvement, such as collection of actual engagement (number of contacts, etc...). Lastly, when it comes to the partner/father, why don't you justify only 1 collection time point for the partner/father? There is the potential for the partner/father to be followed for 40 months during the preconception (12 months), pregnancy (10 months), and postpartum (18 months) periods. Do you not anticipate any changes to the partner/father during that time?

Response: We have provided the justification to the collection of paternal data and clarified the collection timepoints in our responses to Reviewer 2 at #3 and #9. In brief, the collection of baseline paternal information is planned to be used as confounding variables in the analysis, with later data informing a process evaluation. The data to be collected includes paternal lifestyle and anthropometric measures, which will be collected once at each phase of the study period. Given that fathers are not the intervention target group, only limited paternal data will be collected.

Lines 202-204: For the partner, lifestyle and anthropometric measurements will be collected during the preconception, pregnancy, and postpartum periods; blood will be collected once during the pregnancy phase.

Lines 411-414: For the other two co-primary outcomes, linear mixed effects model will be used to examine changes in means between the baseline and follow-up metabolic markers and EPDS scores, with adjustment for baseline potential confounders (including partner's characteristics) and duration of the intervention received.

4. 25% are overweight or obese in Singapore. Do you know the prevalence estimates of overweight and obese independently? Should state if so and justify how you anticipate your outcomes to differ in those having overweight versus those having obesity.

Response: We have updated and provided the proportions of overweight (17%) and obese (13%) women before conception. We anticipated that obese women may have greater improvements in their metabolic and mental health outcomes compared with overweight women.

Lines 95-97: In Singapore, almost one-third of women are overweight or obese [body mass index (BMI) 25-29 kg/m² :17%; BMI ≥30 kg/m² :13%] before pregnancy.

Lines 160-161: We also hypothesize that greater improvements in metabolic and mental health will be observed in women with obesity than those with overweight.

5. Another note is that 25% is a low proportion of the population compared to other countries (US, UK, Australia, for example). Do you really anticipate recruitment of 500 women in preconception time period over 5 years? Especially since, I assume, these women need to be recruited by year 3 to allow for proper follow-up by year 5.

Response: We do expect to complete recruitment of 500 preconception women within the first three years of the trial (based on the experience gained from the S-PRESTO observational cohort which will serve as the comparator arm), with subsequent follow-up of these women through pregnancy and postpartum phases. Recruitment strategies to achieve these targets are stated in Line 216 under Participants and recruitment.

6. Similar to comment above, you plan to recruit 500 women to enroll 400 women in the preconception phase. You anticipate 50% to become pregnant because a previous trial increased pregnancy prevalence, which is a sound rationale. 200 women will be followed throughout pregnancy and 170 through the postpartum period, but at no point do you all talk about drop-out for any reason, especially since you are following these women for such a long period of time. You only account for medical drop out, which is not acceptable. You must speak to the number of anticipated women to not complete the follow-up periods, specifically through the 18-month postpartum period.

Response: We have anticipated a 10% dropout rate through the postpartum period, based on the experience gained from previous birth cohorts GUSTO and S-PRESTO in Singapore, with dropout rates in the first 18-month postpartum about 7-8%.

Lines 438-439: We estimated that 15% of women will experience pregnancy loss and another 10% dropout, leaving 150 dyad pairs to be followed.

7. I do not necessarily agree with your statement: The preconception period represents a unique opportunity where women have the greatest motivation to make a positive change (reference 22). I actually challenge this statement unless further justification may be provided. This statement may only be true in women trying to conceive who are also actively concerned about their pregnancy. Previous evidence suggests that pregnancy is the most potent teachable moment for health behavior change. Not the planning period prior to pregnancy, unless they are actively planning. As such, I believe clarity is needed.

Response: Thank you for the comment. We have amended the statement to add more clarity as to a healthy pregnancy being the underlying motivation during the preconception period.

Lines 116-117: The preconception period represents a unique opportunity where women are motivated to make a positive change to attain optimal pregnancy outcomes.²²

8. You discuss adherence metrics, but do not specify exact primary adherence metric for each time point or period of time. Is there a target behavior you all believe is most important? You should clearly describe your primary and secondary adherence metrics.

Response: All the adherence metrics are given equal importance as we will be tracking them throughout the life-course. We do not have specific metrics to be treated as primary or secondary as those measures are not primary study outcomes.

9. Is there a minimum amount of time the participant must be enrolled in the preconception period for

it to be included in analysis. As an example, if someone were to enroll in preconception and 2 weeks later had a positive pregnancy test, how do you handle their preconception data?

Response: Thank you for bringing up this point. We will censor women who are pregnant within one month from the baseline visit.

Lines 232-233: If a woman is pregnant within one month from the baseline visit, they will be censored from the analyses.

10. At what time point is your criteria for primary and secondary outcomes? You have three (3) distinct time periods, each of critical importance for preconception, pregnancy, and postpartum metabolic and mental health. Is it the 18-month postpartum visit you think is most important? Is so or if not, there needs to be some justification.

Response: Co-primary outcomes of metabolic and mental health will be ascertained at various timepoints throughout the three phases. Indeed, the metabolic and mental health status at the end of the study at 18-month postpartum is the most important, in line with our aim of improving overall metabolic and mental health outcomes.

Lines 359-360: Co-primary outcomes include maternal metabolic health and mental health status in each phase, with 18 months post-delivery serving as the principal endpoint.

11. Your discussion in the subsection "Patients and Public Involvement" about the formative work performed prior to implementation of HELMS should be highlighted much earlier. Citizen and implementation science is extremely important and a key strength of this trial.

Response: Thank you for the suggestion. We have moved the Patient and Public involvement earlier in the manuscript (Line 209).

12. What is the digital platform utilized? A little more information would once again be very helpful.

Response: The intervention will be delivered using a mobile health application.

Lines 52-55: The intervention will be delivered using a mobile health application, to provide anticipatory guidance, raise awareness, and guide goal-setting on lifestyle behaviours that include diet, physical activity, mental wellness, and sleep hygiene from preconception to postpartum.

Lines 243-245: All participants will receive an intervention via a mobile health application, designed to address and improve women's knowledge, attitude, and practice in terms of preconception-pregnancy-postpartum care and health behaviours.

13. No discussion of strengths and limitations.

Response: The strengths and limitations was included under the Article Summary (Line 67), and we have now also added a section to the Discussion.

Lines 530-540: The major strength of this study is the extensive biosampling and detailed phenotyping of mother, father, and child, which will allow longitudinal assessment of changes in health behaviours including diet, physical activity and sleep, maternal metabolic profile, and depression risk. This provides an important platform for biomarker discovery, validation of integrated HELMS interventions, and paves the way for new guidelines for lifestyle programs throughout the life-course. We acknowledge the main limitation of this single-arm trial, where we are unable to distinguish between the effect of intervention, a placebo effect, or the effect of natural history. In addition, HELMS is conducted in a developed country with only English-speaking Asian participants, since English is the language used in our assessment tools and advisories, which may limit its external validity and caution should be exercised before generalizing our findings to other settings and population.

14. Recommend reading other protocol papers for insight into how to improve the current protocol

paper, including, but not limited to, <https://doi.org/10.1186/s13063-021-05204-w>; doi: 10.2196/18211; DOI: 10.1186/s13063-020-4062-4.

Response: Thank you for the comment. We have gained much insights from reading these protocol papers, and we hope our amendments are satisfactory for acceptance for publication.

VERSION 2 – REVIEW

REVIEWER	Sparks, Joshua Pennington Biomedical Research Center
REVIEW RETURNED	07-Nov-2022
GENERAL COMMENTS	I thank the editorial board for the opportunity to re-review the revised manuscript. Additionally, I appreciate the author's responses and revisions to my prior queries and comments. This impactful protocol manuscript for an efficacious life course model approach to improving health behaviors and health in women from the preconception period through pregnancy to postpartum will leave this reviewer excited for the primary outcomes manuscript in the future.